# Nurses’ Techniques for Bottle-Feeding of Infants with Feeding Difficulties: A Qualitative Descriptive Study

**DOI:** 10.3390/nu16213612

**Published:** 2024-10-24

**Authors:** Eri Tashiro, Shingo Ueki, Eri Nagatomo, Junko Miyata

**Affiliations:** 1Department of Health Sciences, Faculty of Medical Sciences, Kyushu University, Fukuoka 812-8582, Japan; tashiro.eri.084@m.kyushu-u.ac.jp (E.T.); nagatomo.eri.018@m.kyushu-u.ac.jp (E.N.); miyata.junko.789@m.kyushu-u.ac.jp (J.M.); 2Department of Pediatric Surgery, Graduate School of Medical Sciences, Kyushu University, Fukuoka 812-8582, Japan

**Keywords:** bottle feeding, feeding difficulties, infant, nurses, qualitative data

## Abstract

**Background/Objectives:** This study identified bottle-feeding techniques for infants with feeding difficulties. **Methods:** Between December 2021 and April 2022, a survey was distributed to nurses with at least 5 years of experience in caring for infants at 1109 hospitals in Japan. The questionnaire included open-ended questions regarding preparation before bottle-feeding, methods of nipple insertion, methods of assisting with sucking, and criteria for continuing bottle-feeding. The responses were classified according to semantic similarity. **Results:** In total, 514 valid responses were received. The respondents had an average of 19.16 years of experience as a nurse or midwife. The most frequently used nipples for infants with feeding difficulties were the Combi Breastfeeding Model, Pigeon Weak Sucking Nipple, and Bean Stalk Nipple. Preparation before bottle-feeding consisted of six categories, including determining the timing of feeding and stimulation to promote wakeup. Nipple insertion methods consisted of four categories, such as assisted opening of the mouth, nipple insertion, and nipple insertion depth. Methods of assisting with sucking were divided into four categories, including encouraging sucking ability from around the mouth and matching the infant’s sucking pace. The criteria for continuing bottle-feeding spanned three categories, including willingness to suck and ensuring necessary nutrition. **Conclusions:** The variety of survey responses describes the current state of bottle-feeding technologies. A combination of several techniques indicated that feeding strategies may lead to effective and appropriate feeding. The nurses continued feeding based on an assessment of the infant’s acceptance. Future studies regarding bottle-feeding techniques must consider the individuality of each infant.

## 1. Introduction

Feeding is the most important practice for a child’s nutritional needs and survival. The primitive sucking reflex enables a healthy child to feed simply by placing the nipple in the mouth. A newborn baby is fed approximately 20 mL of breastmilk or formula eight to ten times a day, with a target weight gain of approximately 30 g per day. An infant should be fed efficiently for over 20–30 min to avoid overusing the infant’s energy and allow for sufficient consumption volume for appropriate weight gain [1,2]. Infants are expected to feed every 2–3 h, which is an important factor in the facilitation of hunger, satiation at the end of a feeding period, digestion, and promotion of the next feeding cycle [2]. However, some infants have difficulty feeding, as they do not have the proper mechanisms for suckling [3,4,5,6]. These infants are often hospitalized for tube or continuous intravenous feeding to avoid malnutrition. Nurses provide bottle-feeding for these infants, although even skilled nurses have difficulty initiating successful bottle-feeding in some infants, resulting in prolonged hospital stays and parental stress and affecting the infant’s growth and development [4,5,7].

Causes of feeding difficulties include morphological abnormalities, such as cleft lip and/or palate [3]; functional abnormalities, such as gastroesophageal reflux disease [4]; neuromuscular disorders, such as severe neonatal asphyxia [5]; and immaturity in preterm infants [6]. Although morphological abnormalities leading to feeding difficulties can be overcome via surgery, other conditions depend on the feeding skills of the nurses. Fucile et al. [6] reported that preterm infants who underwent a pre-feeding oral stimulation program after 48 h of the discontinuation of nasal continuous positive airway pressure achieved independent oral feeding sooner than that observed among those who underwent a placebo stimulation program. Pickler and Reyna [8] suggested that increasing opportunities for bottle-feeding may make it easier to achieve exclusive bottle-feeding. Chen et al. [9] reported that infants who received pre-feeding oral stimulation, non-nutritive sucking, and tactile/kinesthetic interventions transitioned to oral feeding more quickly and gained more weight than that observed with infants who received no intervention or oral stimulation and non-nutritive sucking only. Previous studies regarding the effects of providing stimulation to preterm infants before feeding were reported; however, few studies regarding nurses’ bottle-feeding techniques exist. Even when an infant is awake and willing to suckle, smooth feeding will not be achieved unless the infant achieves a rhythmic sucking, swallowing, and breathing pattern [2,10,11].

Breastfeeding is a commonly used oral feeding method for infants; however, both mothers and infants must learn to move in rhythm [12]. Several suggestions were made to facilitate breastfeeding. The most conducive state for feeding is when the infant is quietly awakened with normal breathing and heart rates [7,12]. The mother should begin breastfeeding by positioning the infant such that his or her head, shoulders, torso, and hips are in a straight line. During breastfeeding, a technique called the dancer hand position is used, in which the mother uses her free hand to support the breast and the infant’s chin, possibly helping the infant stay attached to the breast [7]. Additionally, when effective breastfeeding is not possible, the mother may resort to massaging her breasts while feeding and manually expressing the breastmilk into the infant’s mouth [13]. However, during breastfeeding, milk is only discharged if the infant sucks. When the sucking is weak or inconsistent, more time is required to reach the target feeding volume, leading to increased infant fatigue.

Our previous studies, which reported special bottle-feeding techniques and nipples, characterized bottle-feeding techniques for infants with a cleft lip and/or palate [14]. However, the methods used by nurses for infants with feeding difficulties but without morphological abnormalities are yet to be reported. Therefore, this study used a systematic survey to clarify nurses’ techniques for managing bottle-feeding in infants with feeding difficulties.

## 2. Materials and Methods

### 2.1. Study Design

This study has a qualitative descriptive design. A questionnaire including open-ended items regarding a series of nurses’ techniques during and for preparing, starting, and finishing bottle-feeding and multiple-choice items for feeding bottle types for infants with feeding difficulties was developed by the researchers in this study. This questionnaire was distributed to hospitals in Japan. This study is reported according to the consensus-based checklist for reporting survey studies [15].

### 2.2. Setting and Participants

Hospital File, a website that allows users to search for hospitals throughout Japan, was used on 23 November 2021, to locate hospitals with obstetrics, obstetrics and gynecology, neonatology, or pediatric dentistry departments. This website contained a total of 7964 hospitals, of which 1109 hospitals with the above-mentioned departments were identified. Nurses with experience in caring for infants with general feeding difficulties and with at least 5 years of experience caring for infants with a cleft lip and/or palate were invited to complete the questionnaire to investigate the bottle-feeding techniques used for infants with feeding difficulties but not with a cleft lip and/or palate. The techniques for infants with a cleft lip and/or palate were previously published [14].

### 2.3. Measurements

The questionnaire included items regarding the respondents’ professional qualifications, years of experience, department, and position; the department’s annual number of infants admitted with feeding difficulties other than malformations; and the types of feeding bottles used for infants with feeding difficulties other than malformations. Regarding feeding bottle types, we asked participants to select the type of nipple they used for infants from the list, including the Bean Stalk Beanstalk Nipples, the Combi Breastfeeding Model, and the Medela Cup Feeding. If participants indicated “other” bottles, they were asked to write down the specific bottles. The following open-ended questions regarding any other feeding bottles and bottle-feeding techniques for infants with feeding difficulties other than malformations were also included: (1) “Were there any methods you used to prepare for bottle-feeding?” (preparation before bottle-feeding), (2) “What methods do you use to insert the nipple?” (methods of nipple insertion), (3) “What methods do you use to help the infant suck?” (methods of assisting with sucking), and (4) “What points make you decide whether to continue feeding?” (criteria for continuing bottle-feeding).

### 2.4. Data Collection

Five copies each of the research request form and the questionnaire were mailed to the nursing director at each identified hospital. The nursing directors were informed to distribute copies to nurses or midwives who worked in obstetrics, obstetrics and gynecology, neonatology, or pediatric dentistry for more than 5 years. The respondents’ copy of the request form explained the purpose and methods of the study and requested consent for participation in the anonymous survey. The study targeted 5545 nurses (1109 multiplied by 5). However, because the request for distribution was directed to the nursing director of each hospital, the actual number of questionnaires distributed is unknown.

The data were collected from December 2021 to April 2022. Respondents were required to return a filled questionnaire within 1 month of receiving it. The respondents could print the questionnaire and return it via mail or scan a provided QR code to respond electronically.

### 2.5. Data Analysis

Data regarding the respondent characteristics were analyzed using descriptive statistics. Years of working experience, years of working at their current department, and annual number of infants with feeding difficulties admitted at their department are presented as continuous data (mean ± standard deviation [SD]), while their professional qualification, department, and position and the type of nipple they used for infants with feeding difficulties are presented as categorical data (counts and percentages). Data regarding bottle-feeding techniques were divided into “preparation before bottle-feeding”, “method of nipple insertion”, “method of assisting with sucking”, and “criteria for continuing bottle-feeding” categories prior to analysis. The analysis was conducted using inductive content analysis, which is used when categories are derived from the data [16]. For each scenario, the responses were repeatedly read and scrutinized to fully understand their meaning and were coded into meaning units. Then, units with similar meanings were grouped into subcategories, and subcategories with similar meanings were grouped into categories. Each process was repeated many times and gradually became more abstract. Data analysis was supervised by a pediatric nursing expert with experience in qualitative inductive research. The reliability of the analysis was ensured via multiple discussions with two nurses with experience in the pediatric field.

### 2.6. Ethical Considerations

This study was approved by the Kyushu University Medical School Department Clinical Research Ethics Committee (IRB number: 21077-01). A written request was provided to the nursing director and potential respondents, inviting them to participate in the study. The informed consent form stated that participation in the study was voluntary and that there would be no disadvantage to not participating. The respondents were deemed to have consented to participate in the study by responding to the questionnaire. The survey responses were anonymous, and the respondents were not identified.

## 3. Results

In total, 608 responses (513 by mail and 95 online, response rate: 10.96%) were received; 94 were subsequently excluded as they did not include responses to the open-ended question regarding bottle-feeding techniques. Therefore, 514 responses were included in the analysis (valid response rate: 84.54%). Respondents answered anonymously; thus, we were unaware of which facility or region they worked in.

### 3.1. Respondent Characteristics

The respondent characteristics are shown in Table 1. The mean number of years of experience was 19.16 years (SD = 8.59 years). The respondents worked at their current department for a mean of 10.41 years (SD = 7.17 years) at the time of the survey. The respondents were nearly equally split as nurses and midwives, and most of them worked in the obstetrics and gynecology or neonatology departments. Most respondents were staff nurses. The mean number of infants with feeding difficulties without malformations admitted per year in each department was 13.95 (SD = 31.17). The most frequently used nipples for infants with feeding difficulties (multiple answers allowed) were the Combi Breastfeeding Model (12.65%), Pigeon Weak-Sucking Nipple (12.53%), and Bean Stalk Nipple (11.02%).

### 3.2. Bottle-Feeding Techniques for Infants with Feeding Difficulties

Four aspects of bottle-feeding techniques were analyzed: preparation before bottle-feeding, method of nipple insertion, method of assisting with sucking, and criteria for continuing bottle-feeding. The categories and subcategories of each aspect are shown in Table 2, Table 3, Table 4 and Table 5, respectively. In the following subsections, categories are referred to as [ ], subcategories as < >, raw data as “ ”, and the number of codes as ( ).

#### 3.2.1. Preparation Before Bottle-Feeding

Preparation before bottle-feeding consisted of six categories, 16 subcategories, and 132 codes (Table 2). First, nurses observed the baby’s condition to [determine the timing of feeding]. Nurses [stimulated to promote waking] for sleeping infants, with this category having the highest number of codes (70). The category of [stimulating to promote waking] included the subcategories of <physical stimulation inside and outside the mouth>, including codes of “lip massage” or “tongue massage”; <dripping milk into the mouth>, including codes of “filling the mouth with milk and observing the tongue movement”; <usual care> including the code of “changing diapers to encourage awakening”; and <stimulation to the whole body> including the code of “stimulating the soles of the feet.” The nurses [promoted the crying infants to rest]. <Kangaroo care>, a subcategory of [Promoting to rest], is defined as a position in which the infant rests against the mother’s chest, with skin-to-skin and ventral-to-ventral contact [17]. To prepare for bottle-feeding, the nurses tried to [improve infants’ abdominal bloating] and [improve infants’ physical condition]. The nurses also attempted [preparing for feeding to accept] with subcategories of <preparing the right nipple for each infant> and <adapting the milk for the infant> including codes of “adjusting the milk temperature” and “thickening the milk”.

#### 3.2.2. Methods of Nipple Insertion

The methods of nipple insertion consisted of four categories, 11 subcategories, and 22 codes (Table 3). The nurses considered the depth, placement, and angle of the nipple upon insertion. Subcategories regarding [depth of nipple insertion] included <inserting the nipple deeply> and <shallowly>. Only one respondent answered codes that applied to both the <deeply> and <shallowly> subcategories. Therefore, few nurses reported attempting various depths of nipple insertion. The subcategories of [placement of nipple] included <pressing nipples against tongue> and <not pressing nipples against tongue>, and no nurses reported trying both techniques during a single insertion. No standard [depth of nipple insertion] was reported by 41.08% of the respondents; 61.97% did not have a set method for [placement of nipple], and 70.94% did not have [nipple insertion angle]. Therefore, there is no standardized method for nipple insertion, and the technique is tailored for each infant.

#### 3.2.3. Methods of Assisting with Sucking

The methods of assisting with sucking included four categories, 12 subcategories, and 41 codes (Table 4). The nurses assisted sucking by applying pressure inside the mouth to make sucking easier by [encouraging sucking ability from around the mouth] and helped feed the infant calmly by [matching infants’ sucking pace]. [Encouraging sucking ability from around the mouth] included the subcategories of <supporting the mouth area to make sucking easier>, with codes such as “supporting the lower jaw” or “supporting both cheeks”, and <supporting movement of mouth area in sync with sucking>, with codes such as “lifting the lower jaw in time with sucking”. The respondents also provided [stimulation to encourage sucking] in the infant to continue feeding, similar to the preparation before bottle-feeding. Furthermore, the respondents tried [adjusting to improve feeding], such as <adjusting the milk temperature> or <the nipple> that became crushed during feeding.

#### 3.2.4. Criteria for Continuing Bottle-Feeding

The criteria for continuing bottle-feeding included three categories, 15 subcategories, and 68 codes (Table 5). The respondents attempted bottle-feeding for infants by observing the infant’s condition, including [willingness to suck] and [ensuring necessary nutrition] when the infant first begins to feed. The nurses tried [stabilizing infant’s physical condition] by observing the infant’s vital signs, feeding rhythm, and willingness to suck until feeding is complete.

## 4. Discussion

This study reports a wide variety of detailed feeding techniques used by nurses for infants with feeding difficulties other than malformations. These real-world technologies have immediate potential for use. Prior to bottle-feeding, the nurses stimulated the inside and outside of the infant’s mouth by massaging the mouth and cheeks. This stimulation is not often performed for healthy infants. Fucile et al. [6] demonstrated that oral stimulation programs can advance the maturation of specific sucking skills, which can be further improved with practice. Harding et al. [18] reported that infants who received 10 min of finger/pacifier stimulation three times a day required fewer days to reach full oral feeding. These studies demonstrate that preparation prior to feeding is an effective method to increase infant sucking and feeding volume. Furthermore, regular stimulation of the inside and outside of the mouth prior to oral feeding is effective in facilitating a smooth transition to oral feeding.

At the beginning of a bottle-feeding session, the nurses positioned infants appropriately by positioning their upper body or laying them on one side. When infants are laid on their sides, the head, shoulders, torso, and hips should be aligned in a straight line [7]. This side position results in a slightly greater milk volume than feeding in the upright position, with no difference in physiological stability [19]. The nurses support the infant’s head, which is not yet fully developed, in both feeding positions. When nursing premature infants or infants with certain diseases, several positioning techniques are recommended, and all the positions require head and neck support [20]. The posture is not specific to infants with feeding difficulties. However, as these infants are more likely to vomit due to the immaturity of their swallowing or digestive functions, it is important for nurses to pay attention to key points of posture and position each infant in a way that suits them best to ensure smooth feeding.

To assist infants with sucking, the nurses support the area around their mouth. For breastfeeding, the mother uses her hand to support the breast and the infant’s chin, which may help the infant stay attached to the breast [7]. Hwang et al. [21] reported a feeding method of supporting the infant’s cheek inward and forward with the thumb and ring finger and supporting the infant’s lower jaw and bottle with the little finger, which increased feeding speed and reduced milk leakage during feeding. Hafström et al. [22] reported that sucking was influenced by sex, activity level, and body weight. Based on these results, feeding methods tailored to each infant must be considered to assist in sucking. Therefore, practicing feeding techniques based on the results of the present study may contribute to the selection of appropriate methods that fit the individuality of infants. Future longitudinal studies regarding specific infant feeding techniques are necessary.

During feeding, nurses observed the infants and assessed their vital signs to determine whether the feeding should be continued. Monitoring vital signs can help nurses monitor early signs of clinical deterioration or adverse events [23]. Infants with feeding difficulty may not have stable breathing or proper swallowing techniques; therefore, the child’s sucking and swallowing are not the only determinants of continued feeding. During bottle-feeding attempts, nurses must constantly observe changes in the infant’s vital signs. Additionally, the occurrence of nausea and vomiting must be observed to ensure that the infants receive the necessary nutrition. The transition from tube feeding to bottle-feeding is often not smooth. Infants receiving tube feeding may become hypersensitive to oral stimulation due to the experience of having unknown objects placed into their mouths [24]. Some infants experienced nausea and vomiting during nipple insertion. This reaction may indicate that the infant refuses bottle-feeding [24], and continuing bottle-feeding after this reaction may result in an aversive experience for the infant. Nurses should observe the child’s nipple acceptance and refrain from forcing insertion. In the case of infants with feeding difficulties who show signs of rejection such as nausea, vomiting, and fluctuations in vital signs during feeding, failure to immediately address this condition may lead to deterioration of the infants’ general condition. It is important to perform these observations while recognizing that feeding is a life-threatening activity for these infants.

This study is limited by the open-ended design of the questionnaire, which resulted in abstract or brief responses, making it difficult for researchers to fully understand the meaning of the responses. Furthermore, this study was conducted alongside other surveys [14]. Thus, it is possible that the responses originated from a limited group of respondents with experience in caring for children with a cleft lip and/or palate. Nurses with significant techniques but without experience in caring for children with a cleft lip and/or palate were not targeted in this study. Respondents answered anonymously; thus, there may be facility or regional bias in the responses. In addition, the reasons for feeding difficulties were varied; therefore, it was not possible to accurately understand the infants’ conditions in the context of the respondents’ answers. Therefore, the results of this study did not clarify the feeding techniques that correspond to individual characteristics. This study’s findings reveal that nipple insertion depth, positioning, and angle vary depending on each infant. However, some cases might exist in which feeding cannot be improved using these techniques. In cases where children have dysphagia or where there are problems with the parents, families, or environment, other approaches would be needed to improve the nutrition status of the child.

Future studies regarding specific feeding techniques for infants with specific characteristics and conditions must be conducted to establish more effective methods of assistance while considering the infants’ individual characteristics, feeding environment, and feeding person. The findings of this study can form a basis for further research.

## 5. Conclusions

The varied responses to the survey regarding bottle-feeding techniques for infants with feeding difficulties other than malformations indicate the current state of bottle-feeding technologies. The combined use of several techniques may lead to effective and appropriate feedings. The nurses assess the infants’ acceptance when deciding to continue bottle-feeding attempts. However, as no detailed information regarding the infants’ conditions was reported, many of the answers were abstract or brief, and it was difficult to determine the appropriate feeding techniques for each infant’s individual needs. In addition, the feeding environment and feeding person were not investigated. Thus, the impact of these factors is unknown. The results of this study reveal that nipple insertion depth, positioning, and angle vary depending on each infant. Future studies should consider examining bottle-feeding techniques based on the specific characteristics or conditions of each infant.

## Figures and Tables

**Table 1 nutrients-16-03612-t001:** Respondent characteristics (N = 514).

Variable	N	%
**Professional qualifications**		
Nurse	277	53.90
Midwifery	230	44.75
Other	4	0.77
Unanswered	3	0.58
**Department**		
Obstetrics and gynecology	237	46.11
Neonatology	233	45.33
Pediatrics	21	4.09
Pediatric dentistry	7	1.36
Other	11	2.14
Unanswered	5	0.97
**Position**		
Staff nurse	404	78.60
Deputy chief nurse	82	15.95
Chief nurse	13	2.53
Director of nursing/Deputy director of nursing	4	0.78
Unanswered	11	2.14

**Table 2 nutrients-16-03612-t002:** Categories and subcategories of preparation before bottle-feeding.

Categories	Subcategories
Determining the timing of feeding	Observing the condition (2)
	Adapting to infant’s movements (9)
Stimulating to promote waking	Physical stimulation inside and outside the mouth (39)
	Dripping milk into the mouth (5)
	Usual care (12)
	Stimulation to the whole body (14)
Promoting to rest	Calm down (8)
	Kangaroo Care (1)
Improving infant’s abdominal bloating	Promoting abdominal decompression (12)
	Observing abdominal symptoms (4)
	Promoting stomach decompression (5)
Improving infant’s physical condition	Raising upper body (3)
	Improving posture (7)
	Observing physical condition (5)
Preparing for feeding to accept	Preparing the right nipple for each infant (4)
	Adapting the milk for the infant (2)

Note: The number of codes is shown in parentheses.

**Table 3 nutrients-16-03612-t003:** Categories and subcategories of methods of nipple insertion.

Categories	Subcategories
Assisted opening the mouth and nipple insertion	Using nurse’s fingers to help open the mouth (3)
	Slipping a nipple into infant’s mouth (2)
Depth of nipple insertion	Inserting the nipple until it sets on infant’s tongue (6)
	Inserting the nipple deeply (2)
	Inserting the nipple shallowly (1)
Placement of nipple	Pressing nipples against tongue (1)
	Not pressing nipples against tongue (1)
	Making close contact with the nipple against the lips (2)
Nipple insertion angle	Placing nipple against the palate (1)
	Paralleling to the tongue (2)
	Adjusting the bottle angle to prevent too much milk (1)

Note: The number of codes is shown in parentheses.

**Table 4 nutrients-16-03612-t004:** Categories and subcategories of methods of assisting with sucking.

Categories	Subcategories
Encouraging sucking ability from around the mouth	Supporting the mouth area to make sucking easier (3)
	Supporting movement of mouth area in sync with sucking (2)
Matching infant’s sucking pace	Matching infant’s sucking (1)
	Matching infant’s pace (1)
	Regulating breathing (2)
Stimulating to encourage sucking	Adjusting nipple insertion (4)
	Stimulating in the oral cavity to encourage sucking (6)
	Stimulating the mouth area to encourage sucking (5)
	Stimulating in sync with tongue movement to encourage sucking (5)
	Stimulating the body to promote awake (9)
	Tapping the bottle to encourage sucking (1)
Adjusting to improve feeding	Adjusting the milk temperature (1)
	Adjusting the nipple (1)

Note: The number of codes is shown in parentheses.

**Table 5 nutrients-16-03612-t005:** Categories and subcategories of criteria for continuing bottle-feeding.

Categories	Subcategories
Willingness to suck	Awaken (2)
	Moving activity (5)
	Continuing sucking (5)
	Sucking naturally (12)
	Accepting nipples (14)
	Reacting to stimuli (8)
	Continue feeding after the nipple is removed (3)
	Resuming sucking (4)
Ensuring necessary nutrition	Swallowing without spilling (4)
	Progressing feeding smoothly (3)
	Not nauseated or vomiting (2)
Stabilizing infant’s physical condition	Stable vital signs (2)
	Not choking (1)
	Good feeding rhythm (3)

Note: The number of codes is shown in parentheses.

## Data Availability

The data presented in this study are available upon request from the corresponding author. The data are not publicly available due to restrictions on the ethical considerations for each participant.

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
