# Peer review of "Nurses’ Techniques for Bottle-Feeding of Infants with Feeding Difficulties: A Qualitative Descriptive Study"

_nutrients, 2024, doi:10.3390/nu16213612_

Round 1

Reviewer 1 Report

Comments and Suggestions for Authors

Dear authors,

Thank you very much for creating this work on a reasonably interesting topic for nurses, midwives, paediatricians, etc …

I have included some comments and suggestions to improve the quality and readability of your work.

Wish you all the best in the re-submission process

Best regards

Reviewer  

Page 2. Line 94.

Can the authors clarify the total numbers of hospitals in Japan? They included 1,109 hospitals in this survey?

Page 2. Line 96.

Can the authors elaborate on the total numbers of nurses they have approached for this questionnaire? What was the percentage of responders?

Page 3. Line 98.

The nurses in this study had large experience of more than 5 year in caring and feeding assistance for children with cleft. However, infants in this survey had no cleft of potentially had their cleft repaired already. Could this be a bias in collecting and interpretation of data. If yes, this should be highlighted in the limitation.

The general impression from this survey, at least to my opinion, is a very strong association between the feeding difficulties and bottle feeding techniques. However, bottle feeding techniques and challenges are not the only reason for a child who feeds poorly. Many other conditions such as child’s, parents’, and environmental situations could also be involved in this problem.

The authors should consider to express this in the introduction, discussion and conclusion sections. They also should consider to mention that they have only approached one of these many reasons in this survey.

Comments on the Quality of English Language

No comments.

Author Response

For reviewer #1

We would like to thank you for your many detailed comments. The responses to each comment are shown below.

Page 2. Line 94.

Can the authors clarify the total numbers of hospitals in Japan? They included 1,109 hospitals in this survey?

Response: On page 2, lines 94-96, I have added the following:

This website contained a total of 7,964 hospitals, of which 1,109 hospitals with the above-mentioned departments were identified.

Page 2. Line 96.

Can the authors elaborate on the total numbers of nurses they have approached for this questionnaire? What was the percentage of responders?

Response: On page 3, lines 123-126, we have added that the survey targeted 5,545 nurses. The response and valid response rates have also been added on page 4, lines 160 and 163.

Page 3. Line 98.

The nurses in this study had large experience of more than 5 year in caring and feeding assistance for children with cleft. However, infants in this survey had no cleft or potentially had their cleft repaired already. Could this be a bias in collecting and interpretation of data. If yes, this should be highlighted in the limitation.

Response: As you pointed out, the participation criteria were limited. However, the survey’s questions were divided into two categories: to children with and without cleft lip and/or palate. So, the study’s results included only techniques for children with feeding difficulty and without cleft lip and palate. Considering the limited criteria for participation, the techniques of nurses without experience in caring for children with cleft lip and/or palate were not included. This was added as a limitation:

In line 304-308:

Furthermore, this study was conducted alongside other surveys [14]. Thus, it is possible that the responses originated from a limited group of respondents with experience in caring for children with cleft lip and/or palate. Nurses with significant techniques but without experience in caring for children with cleft lip and/or palate were not targeted in this study.

The general impression from this survey, at least to my opinion, is a very strong association between the feeding difficulties and bottle feeding techniques. However, bottle feeding techniques and challenges are not the only reason for a child who feeds poorly. Many other conditions such as child’s, parents’, and environmental situations could also be involved in this problem.

The authors should consider to express this in the introduction, discussion and conclusion sections. They also should consider to mention that they have only approached one of these many reasons in this survey.

Response: As you suggested, some factors that cause feeding difficulties cannot be solved by nurses’ techniques alone. I have added the following as a research limitation.

In lines 314-321:

However, some cases might exist in which feeding cannot be improved using these techniques. In cases where children have dysphagia or where there are problems with the parents, families, or environment, other approaches would be needed to improve the nutrition status of the child.

Future studies regarding specific feeding techniques for infants with specific characteristics and conditions must be conducted to establish more effective methods of assistance while considering the infants’ individual characteristics, feeding environment, and feeding person. The findings of this study can form a basis for further research.

In line 329-331:

In addition, the feeding environment and feeding person were not investigated. Thus, the impact of these factors is unknown.

Reviewer 2 Report

Comments and Suggestions for Authors

OVERVIEW

This paper reports a qualitative study of bottle-feeding techniques for infants with feeding difficulties at hospitals in Japan.

POSSIBLE MAJOR ISSUES

My impression reading this paper is that it would be much more informative if there was more quantitative information provided. As it stands, it provides a list of all the techniques which the respondents reported using. Would it not be more informative to say how often each category and subcategory of feeding technique was reported by respondents?

MINOR ISSUES

Responses come from 514 respondents at 1,109 hospitals. However, it’s not made clear whether these 514 respondents are at different hospitals, or whether in some instances there may have been more than one respondent at a single hospital.

It would be helpful if terms that might not be familiar to readers were explained. For example, “kangaroo care”.

Author Response

For reviewer #2

Thank you for your careful review. Below, we have provided responses to each of your comments. We hope that we have answered all of your questions.

Comments and Suggestions for Authors

POSSIBLE MAJOR ISSUES

My impression reading this paper is that it would be much more informative if there was more quantitative information provided. As it stands, it provides a list of all the techniques which the respondents reported using. Would it not be more informative to say how often each category and subcategory of feeding technique was reported by respondents?

Response: I added the number of codes obtained to clarify the categories and subcategories in Tables 2–5. On page 5, line 185, I have also added how the numbers are presented.

MINOR ISSUES

Responses come from 514 respondents at 1,109 hospitals. However, it’s not made clear whether these 514 respondents are at different hospitals, or whether in some instances there may have been more than one respondent at a single hospital.

Response: In the “Results” section on page 4, lines 163-164, we have added that we do not know the facility or region where the respondents worked, and in the “Discussion” section on page 8, line 308-309, we have suggested the possibility of bias due to facility or region as a limitation of the study.

It would be helpful if terms that might not be familiar to readers were explained. For example, “kangaroo care”.

Response: Thank you for your comment. On page 5, lines 196-198, we have added an explanation of Kangaroo Care.